# INTERPRETABLE CONCEPT DISCOVERY AND LEARNING FROM PRETRAINED VISION-LANGUAGE MODELS

## ABSTRACT

Vision-language models (VLMs) pretrained on web-scale data excel at recognizing objects and scenes. However, it remains mysterious if and how the VLMs learn and utilize rich semantic information of visual concepts, such as colors and shapes, for recognition. While some prior work concluded that pretrained VLMs do not capture interpretable concepts, other work observed that leveraging the concept-based text prompts improves visual recognition accuracy, and appears to offer some degree of interpretability. In this paper, we aim to address this discrepancy and understand pretrained VLMs' true capability of encoding interpretable visual concepts. We identify that the different strategies of concept selection and concept prompting lead to different conclusions from prior works, and that when class names are included in the concept prompts, the resulting activations are highly useful for classification, but often unrelated to the prompted concept. To address these challenges, we propose a new framework to jointly discover and learn interpretable visual concepts from pretrained VLMs. Our discovered concepts are class-agnostic, and selected based on the visual discriminability measured by mutual information between images and concepts. We then propose a self-supervised framework to adapt the pretrained VLM to recognize the discovered concepts with higher interpretability. Through extensive quantitative and human evaluations, we demonstrate that our concept discovery and learning (CDL) framework significantly improves the interpretability of the discovered concepts, while achieving state-of-the-art performance on concept-based visual recognition. All code and data related to this paper will be made public.

## 1 INTRODUCTION

Vision-and-language models (VLMs) such as CLIP (Radford et al., 2021) can perform accurate image classification tasks in zero-shot setting by leveraging language prompts such as "a photo of a name-of-class". However, this paradigm is not interpretable since the prompts of class names do not contain visual information and the reasoning clues are unclear. In order to exploit the rich semantic information contained by the target class, researchers (Pratt et al., 2023; Menon & Vondrick, 2022) have proposed to generate descriptors for class names with external knowledge bases such as large language models (LLMs). For example, given the question "How to visually describe pelican?", an LLM may generate descriptors like "A pelican is a large bird with white plumage, a long, curved bill, and large webbed feet". Those descriptors visually describe specific classes of objects and may provide better performance served as prompts for VLMs on zero-shot recognition tasks, since they contain visual concepts as reasoning clues.

Our paper first asks the question: Do these concept-augmented text prompts actually offer interpretability? As illustrated in Figure 1, we observe that the role of concepts in the LLM-generated descriptors is insignificant. We take VDES (Menon & Vondrick, 2022), an concept-augmented prompting method as an example: if we mask out the class names from the descriptors, the zero-shot classification accuracy of the CLIP model drops catastrophically (see Table 7). When the concepts are randomly shuffled but the class names are unchanged, the classification accuracy remains unchanged but the retrieved concepts are uncorrelated with the class names. These observations demonstrate that the concept-enhanced text prompts, which condition the concept selection on specific class names, do not provide interpretability for the resulting zero-shot classifier, despite achieving higher classification accuracy.

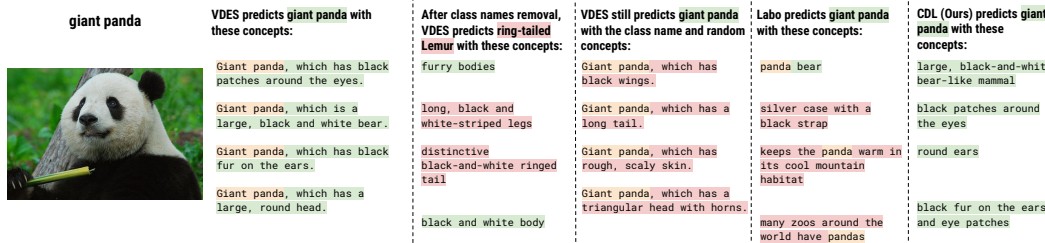

Figure 1: Examples of how different models conduct image classification based on the concepts. Correct predictions and concepts are in green, while wrong concepts and non-visual concepts are in red. Though VDES (Menon & Vondrick, 2022) and LaBo (Yang et al., 2023b) can both classify the image correctly and the concepts are mostly correlated with *giant panda* (highlighted in orange), most concepts directly contain the class name "giant panda". After the removal of class name in VDES, we observe that VDES classifies this image as ring tailed lemus and correlate the image with irrelevant concepts. In addition, even with random concepts VDES can still predict *giant panda* correctly based on the class name in the prompts. Our proposed method (CDL) can predict *giant panda* correctly based on the class-agnostic concepts.

We then ask the question: How can we utilize the class-agnostic concepts to decompose the reasoning of VLMs? The previous example shows that the zero-shot classification capacity of VLMs does not work with class-agnostic concepts. Hence, to achieve both accurate and interpretable classification, it is necessary to learn a concept-class map under supervision. The Concept Bottleneck Model (CBM) (Koh et al., 2020) is a method to decompose the end-to-end recognition of vision-language models into concept-level predictions, which trains a linear classifier based on the model's predicted similarity between the images and concepts. Previous work LaBo (Yang et al., 2023b) proposes to train CBMs with automatically discovered concepts from LLM-generated descriptors, and it achieves competitive clssification performance. However, the learned concept-class mapping is often not factual and groundable, and thus not interpretable according to quantitative and human evaluations. We observe that concepts discovered by LaBo contain many non-visual and class-biased concepts, which may not serve as reasoning clues for visual recognition. Additionally, previous research (Yun et al., 2023; Lewis et al., 2022) shows that the concepts directly recognized by a pretrained VLM (e.g. CLIP) are often noisy. In order to build interpretable multi-modal recognition paradigm, we need to **discover** a set of class-agnostic and discriminative concepts, and **learn** the concepts by adapting a pretrained VLM to better recognize the discovered concepts.

For concept discovery, we propose to query LLM with more visual-focused prompts and extract a list of general and class-agnostic visual concepts from the generated descriptors. We then design a Mutual Information-based method to evaluate the discriminability of concepts and select a discriminative and expressive concept set. With respect to concept learning, we propose to fine-tune the pre-trained CLIP model to learn visual concepts in a self-supervised way. We turn the image-text matching task of the CLIP model into the classification objective with a fixed concept-to-class weight matrix initialized with LLM knowledge. To avoid extra supervision, we generate the class labels by matching the images with class names using CLIP. Inspired by previous work (Kirichenko et al., 2022) that shows last-layer fine-tuning is enough to map strong vision encoders to correct concepts, we only fine-tune the projection layers of vision and text encoders of the CLIP model. In experiments, both part of our methods significantly improve the performance and interpretability of the CBM-based multi-modal recognition according to both automatic and human evaluated metrics.

To conclude, we make the following contributions: First, we investigate the discrepancies of prior work on whether pretrained VLMs encode interpretable visual concepts, by closely inspecting the VLM-based concepts for visual recognition. We reveal that the discrepancies are due to different concept discovery mechanism, and the class-biased concept prompting to the VLM. Second, we propose a simple and effective approach to automatically discover class-agnostic visual concepts and build general and extensive concept dictionaries. We also propose an efficient self-supervised method to adapt a pretrained VLM to recognize the visual concepts with higher interpretability. Finally, we conduct comprehensive experiments including human evaluations with respect to the interpretability of multi-modal recognition. Our model demonstrates the state-of-the-art performance on both recognition accuracy and interpretability.

## 2 RELATED WORK

### 2.1 MULTI-MODAL RECOGNITION

Vision-and-language models (VLMs) pretrained on unlabeled pairs of images and texts from the internet have shown great success on multimodal benchmarks. Representations learned by these VLMs can be transferred to a wide range of tasks, such as visual question answering (Li et al., 2020; 2022; Bai et al., 2023), and image and video captioning (Lu et al., 2019; Zhang et al., 2021; Yang et al., 2023a). These pretrained VLMs can directly recognize complex concepts in a zero-shot setting, such as object categories with text prompts (Alayrac et al., 2022; Radford et al., 2021). However, it remains unclear whether VLMs learns to utilize rich information of visual concepts (e.g. colors and textures) for such zero-shot capability. Previous works (Pratt et al., 2023; Menon & Vondrick, 2022) propose to introduce visual concept knowledge into the multi-modal recognition process through querying large language models (LLMs). In this paper we illustrate the limitation of those class-conditioned prompts and propose a simple yet effective method to automatically discover concepts which can be applied to VLMs.

### 2.2 EMERGENCE OF CONCEPTS IN LANGUAGE MODELS

Leveraging Transformer architecture (Vaswani et al., 2017), large language models (LLMs) have shown its impressive capability on various tasks, such as code writing (Zhang et al., 2023), mathematical problem solving (Lewkowycz et al., 2022), question answering (Sanh et al., 2021; Wei et al., 2021). However, some argue that solely training on word symbols will not lead to the emergence of understanding of the concepts and meanings of words (Bender & Koller, 2020). On the contrary, recent works find evidence that language models can understand simple concepts, such as colors and spatial directions (Patel & Pavlick, 2022; Li et al., 2023).

### 2.3 CONCEPT DISCOVERY AND LEARNING

The utilization of visual concepts such as colors and shapes can benefit various downstream tasks in computer vision and multi-modal learning, including object detection (Yao et al., 2022) and visual question answering (Mao et al., 2019). Concept Bottleneck Model (CBM) is a method to explicitly decompose the stage of composite concept inference with primitive visual concepts (Koh et al., 2020). The interpretability and performance of CBM have been investigated in (Havasi et al., 2022; Leemann et al., 2023; Moayeri et al., 2023). Previous work introduces CBM to VLM-based image classification with concepts generated by LLMs and observe significant performance boost on image classification tasks (Yang et al., 2023b; Yan et al., 2023). However, when applying intervention approach (Koh et al., 2020) to the trained CBMs, the classification performance drops, which reflects that the model do not learn the visual concepts sufficiently (Yun et al., 2023). In this work, we further analyze the interpretability of VLM-based zero-shot classification and CBMs.

## 3 METHODOLOGY

In this section, we first introduce the background of CLIP-based multi-modal recognition paradigm and Concept Bottleneck Model (CBM) (in Sec. 3.1). After that, we detail our methods to build a class-agnostic visual concept dictionary from LLM-generated descriptors and select discriminative concepts for classification (in Sec. 3.2). Then we illustrate out methods to fine-tune CLIP to better learn visual concepts with self-generated supervision (in Sec. 3.3).

### 3.1 BACKGROUND

**CLIP-based Multi-modal Recognition.** CLIP (Radford et al., 2021) is a vision-language model that contains an image encoder and text encoder. CLIP learns to align images and texts in the embedding space during pre-training. The pre-trained image and text encoders can conduct unsupervised multi-modal recognition through the following paradigm. Given a set of categories $\mathbb{C}$, we first generate a set of text prompts $\mathbb{P}$, which involves text prompts of "A photo of a/an $c$", for each class $c \in$

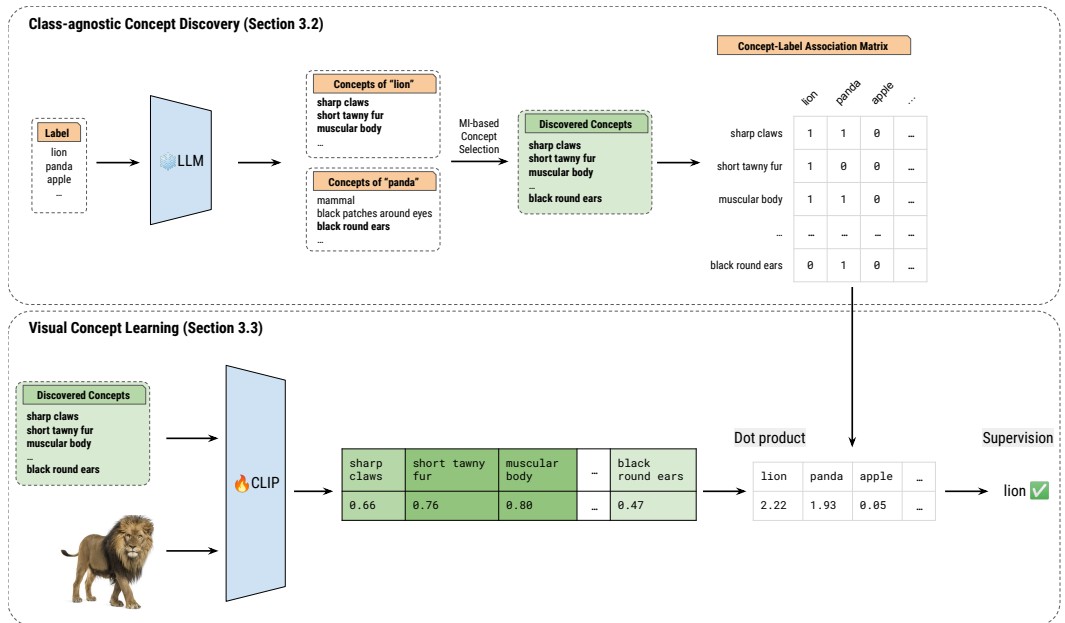

Figure 2: Illustration of our proposed concept discovery method and concept learning architecture. For concept discovery, given the set of class labels, we first utilize a frozen LLM to discover a list of class-agnostic concepts and carry out mutual-information-based concept selection to filter these concepts. We process the discovered concepts into a concept-label association matrix. For concept learning, we use CLIP (Radford et al., 2021) to compute a vector of alignment scores between an image and each concept. To classify this image, we take dot product between the alignment vector and the association matrix. During concept learning, we train the last projection layer in CLIP.

$\mathbb{C}$. For a given image $x$, the label $y \in \mathbb{C}$ can be predicted through finding the most similar prompts:

$$y = \arg\max_{i} \text{Cossim}(\mathbf{E_I}(x), \mathbf{E_T}(\mathbb{P}_i)), \tag{1}$$

where $\mathbf{E_I}$ denotes the image encoder and $\mathbf{E_T}$ denotes the text encoder and $\text{Cossim}$ means the cosine similarity between the image and text embeddings. In the previous research and our work, the text prompts can be replaced by the LLM-generated descriptors and the discovered concepts.

**Concept Bottleneck Model.** Concept Bottleneck Model (CBM) (Koh et al., 2020) proposes to learn a direct projection from concepts to categories to analyze the decision basis of visual recognition models. Given a set of concepts $\mathbb{P}$ and a set of categories $\mathbb{C}$, it learns a $|\mathbb{P}| \times |\mathbb{C}|$ matrix W to project the concept space to the category space. For an input image x, we first calculate the similarity scores between the image and each concept p in $\mathbb{P}$ with CLIP and then feed image-concept similarity scores into the CBM W to get the predicted label $y \in \mathbb{C}$ as shown in the following equation:

$$y = \arg\max \mathbf{E_I}(x) \cdot \mathbf{E_T}(\mathbb{P})^T \cdot W \tag{2}$$

### 3.2 CLASS-AGNOSTIC CONCEPT DISCOVERY AND SELECTION

Previous works (Menon & Vondrick, 2022; Pratt et al., 2023; Yang et al., 2023b) propose various methods to generate descriptive features through querying LLMs with designed prompts like "How to describe a {class name} in a photo?". However, the generated descriptions cannot provide enough interpretability for image classification since they are class-dependent and not visually discriminative, which is illustrated by the examples in Figure 1 and experiments in Sec. 4. In our work, in order to focus on generating visually descriptive features, we query the LLM with the prompt "What are useful visual features for distinguishing a {category name} in a photo?" and in-context examples.

Through prompting the LLM, we can obtain a visual concept set $\mathbb{S}$ and a class-concept map M. If concept $j$ is contained by the descriptors of the category $i$, $M_{ij}$ is 1, and otherwise $M_{ij}$ is 0. Previous

work (Yan et al., 2023) illustrate that in order to build a interpretable decision basis of CBM, it is important to select an expressive subset containing a small number of discovered concepts. To select important and discriminative concepts, we introduce Mutual Information (MI) to evaluate the discriminability of the concepts. MI measures the mutual dependence of two variables $X \in \mathbb{X}$ and $Y \in \mathbb{Y}$ with the following equation:

$$MI(X, Y) = \sum_{y \in \mathbb{Y}} \sum_{x \in \mathbb{X}} P_{X,Y}(x, y) \log \frac{P_{X,Y}(x, y)}{P_X(x) P_Y(y)} \tag{3}$$

Given a concept $s \in \mathbb{S}$ and a set of $k$ images and their coresponding categories, we define the $k$-dimensional vectors $X$ and $Y$, where $X_k$ is the cosine similarity between the concept and the $k$-th image and $Y_k$ is the bool value whether the concept is contained by the categories of the images, which can be retrieved from the class-concept map $M$. The Mutual Information between $X$ and $Y$ evaluates the mutual dependency between image-concept similarity and the "ground-truth" [1] concept containing label of the images. Hence, high MI score indicates that the concept has high similarity with images it describes and has low similarity with images that do not contain it, which means that the concept is discriminative and can provide useful information for the image classification. Therefore, we select top-$k$ concepts with the highest MI scores to build the concept dictionary for image classification.

In order to diversify the concept list, we utilize the cosine similarity of CLIP text embeddings to filter duplicate concepts. We define a similarity threshold $th$, if the similarity of two concepts is higher than $th$, we delete the concept with low MI score.

### 3.3 Visual Concept Learning

Previous works (Yun et al., 2023; Lewis et al., 2022) show that the image-text pair based contrastive learning cannot endow VLMs the ability to directly predict or bind primitive visual concepts. Meanwhile, previous research illustrates that pre-trained VLMs can learn useful visual patterns and embed visual concept knowledge in their architectures. Therefore, exploiting the powerful vision and text encoders of the CLIP model, we propose to re-align the learned visual patterns with textual concepts through fine-tuning their last projection layers. We also design a novel self-supervised learning architecture to leverage the image-category alignment knowledge in the CLIP model to teach itself to learn concepts.

**Concept Learning Architecture**   In order to utilize the image-category alignment knowledge of CLIP, we propose to turn the concept learning into a classification objective with the help of CBM, which maps concepts to corresponding categories. For a given image, we conduct the dot product between the image embedding and text embedding to get the image-concept similarity, and then we pass the image-concept similarity into the CBM weight matrix $W$ fixed by the concept-class mapping $M$ in Sec. 3.2 to get the class-label prediction.

**Self-supervised Learning Paradigm**   Instead of using the human-labeled data, we propose to leverage the knowledge inside CLIP to fine-tune itself. We generate the class labels using the unsupervised recognition paradigm in Sec. 3.1 and the LLM-generated descriptors with class names. Although containing noise, the CLIP predicted labels are of satisfactory accuracy and can provide correct supervision for concept learning. Since we select concepts with image labels in the concept discovery stage, we conduct the fine-tuning with all discovered concepts instead of the selected concepts to avoid introducing of extra supervision.

## 4 Limitation of previous works

In this section, we design experiments to reveal the limitations of previous works in concept-based multi-modal recognition. We first illustrate the LLM-generated descriptors in Menon & Vondrick (2022) might be biased by class names. Then we showcase the existence of non-visual and class-biased concepts and their effect to the multi-modal recognition performance.

---

[1] the labels come from LLM knowledge

|                              | ImageNet | Food-101 | CIFAR-100 | CIFAR-10 | CUB-200 | Flowers-102 |
|------------------------------|----------|----------|-----------|----------|---------|-------------|
| CLIP + Name                  | 71.6     | 91.8     | 75.9      | 96.2     | 63.1    | 77.4        |
| CLIP + Name w/ Concept       | 75.0     | **92.4** | 77.7      | **96.6** | 63.5    | 78.9        |
| CLIP + Concept               | 22.1     | 3.6      | 30.9      | 70.7     | 5.3     | 7.0         |
| CLIP + Name w/ Random Concept| 70.1     | 91.6     | 75.4      | 95.0     | 62.1    | 78.7        |
| CDL + Concept                | **75.7** | 91.5     | **77.8**  | 96.5     | **64.7**| **80.9**    |

Table 1: The unsupervised classification results of the original and our fine-tuned CLIP model with different prompts. "Name" corresponds to the simple prompt "A photo of a class name". "Name w/ Concept" denotes the prompts in the previous work (Menon & Vondrick, 2022), which are like "A photo of a class name, which has "concept". "Concept" corresponds to the pure concept. "Name w/ Random Concept" means that we replace the correct concept with random concepts. "CDL + Concept" means the prediction of our fine-tuned CLIP model with class-agnostic concepts.

### 4.1 CLASS-CONDITIONED DESCRIPTORS

Previous work (Menon & Vondrick, 2022) proposed to utilize LLM to obtain descriptive features for categories, and then conduct recognition based on the similarity between the descriptive features and images. However, the concepts in previous work might be biased by class names. We design the experiment to remove the class name in the prompts and compare the zero-shot classification accuracy of the CLIP model on different types of prompts. As shown in Table 7, we can observe that the zero-shot classification accuracy of the CLIP model drops catastrophically when removing the class name ("CLIP + Concept" row). In addition, we also randomly shuffle the descriptions and keep the class names in the prompts. From the results of "CLIP + Name w/ Random Concept" row we can see that even if the descriptors are randomly shuffled the CLIP model can correctly match most images with the class names in the prompts. This phenomenon demonstrates that descriptors themselves cannot bring satisfactory recognition performance because that the class names make a decisive difference in the recognition results. Thus, it is hard to draw conclusion that LLM-generated class-condition prompts can provide enough interpretability. The results of "CDL + Concept" row shows that our fine-tuned CLIP model can achieve comparable performance with VDES while we do not include any class-related information in our concepts, which illustrates that our model can learn correct associations between concepts and classes and provide concept-level interpretability.

### 4.2 NON-VISUAL CONCEPTS

As shown in the examples, the discovered concepts in LaBo (Yang et al., 2023b) contain a lot of visually non-discriminative concepts such as "found in North America" and "a magnificent animal", which cannot provide visual clues to interpret the decision of the model. In addition, many of LaBo concepts are also biased by class names. We conduct a human evaluation to compare our discovered concepts with LaBo on the proportion of non-visual concepts and concepts containing class names. We first use the concept selection methods of our work and LaBo to select 400 concepts in the CUB dataset and let human annotators to evaluate them. The details of our human evaluation can be found in the appendix. According to the human evaluation, **36.50**% of LaBo concepts are not visually discriminative while only **19.75**% of our discovered concepts are not visually discriminative. **33.25**% of LaBo selected concepts contain class name information such as "one of the largest altrobass" for the class "black-footed altrobass", while only **8.75**% of our discovered concepts contain class name information. The results illustrate that our discovered concept are much more visually discriminative and class agnostic compared to LaBo. Hence, our concepts can serve as better reasoning clues and provide more interpretability for multi-modal recognition.

## 5 EXPERIMENTS

### 5.1 EXPERIMENTAL SETUP

**Datasets** We conduct experiments on several general and fine-grained image classification datasets including ImageNet (Deng et al., 2009), Food-101 (Bossard et al., 2014), CIFAR-100 (Krizhevsky et al., 2009), CIFAR-10, CUB-200 (Wah et al., 2011) and Flowers-102 (Nilsback & Zisserman, 2008). The statistics of the datasets are shown in the appendix.

|  | ImageNet | | Food-101 | | CIFAR-100 | | CIFAR-10 | | CUB-200 | | Flowers-102 | |
|---|---|---|---|---|---|---|---|---|---|---|---|---|
| #Concepts | 1000 | 2000 | 101 | 202 | 100 | 200 | 10 | 20 | 200 | 400 | 102 | 204 |
| CLIP+LaBo | 83.2 | 83.6 | 89.8 | 91.1 | 80.5 | 84.1 | 77.8 | 92.2 | 79.7 | 81.3 | 95.5 | 94.9 |
| CLIP+CDL | 83.6 | 83.7 | 94.3 | 94.8 | 83.2 | 85.1 | 80.9 | 92.6 | **83.2** | **83.4** | 96.3 | 95.7 |
| CDL+CDL | **83.8** | **83.9** | **94.4** | **94.9** | **83.6** | **85.3** | **96.1** | **96.5** | 82.5 | 82.1 | **96.6** | **96.2** |

Table 2: Comparison with LaBo on classification accuracy with different bottleneck size.

|  | ImageNet* | | Food-101 | | CIFAR-100 | | CIFAR-10 | | CUB-200 | | Flowers-102 | |
|---|---|---|---|---|---|---|---|---|---|---|---|---|
| #Concepts | 397 | 794 | 101 | 202 | 100 | 200 | 10 | 20 | 200 | 400 | 102 | 204 |
| LM4CV | **75.7** | 75.8 | 80.2 | 81.9 | **75.1** | 77.3 | 80.1 | 88.0 | 63.9 | 64.1 | 87.3 | 89.0 |
| CDL | 75.4 | 75.8 | **85.9** | **86.9** | 74.9 | **77.6** | **85.6** | **89.4** | **67.3** | **69.6** | **88.7** | **89.2** |

Table 3: Comparison with LM4CV (Yan et al., 2023) on classification accuracy with different bottleneck size. For ImageNet, we use the same ImageNet-Animal subset with LM4CV.

**Baselines** We compare with LaBo (Yang et al., 2023b) and LM4CV (Yan et al., 2023), which is the state-of-the-art works in CBM-based image classification. Following the setting in LM4CV, we control the bottleneck size (number of concepts) to be the same for baselines and our model for fair comparison. For LM4CV, we can only compare the classification performance with the reported results since we cannot access the code and data of this work.

**Implementation Details** We also use the same LLM (GPT-3-text-davinci-002) to obtain descriptors and the same CLIP backbone (Vit-L-14 to compare with LaBo and Vit-B-32 to compare with LM4CV) for image and text encoding. For concept discovery, we utilize the sklearn toolkit "Mutual Information Regression" to calculate the Mutual Information of continuous values by entropy estimation from k-nearest neighbors. The similarity threshold to filter duplicate concepts is set to 0.9. Lower threshold will select less concepts and higher threshold will include more similar concepts. We showcase the performance of different threshold in the Appendix. Following Yun et al. (2023), we use logistic regression to train the concept bottleneck models. We used the default sklearn hyperparameters for all datasets. We observe that the performance of CBM is robust to the choice of hyperparameters. For concept learning, we use the AdamW optimizer with 5e-4 learning rate and 1e-4 weight decay to fine-tune the CLIP model, and we use the validation loss to select checkpoints. More implementation details can be referred in our attached code. For human evaluation experiments, we hire workers from Amazon Mechanical Turk. For each data we ask three human worker to annotate and use the majority vote to obtain the result. More details about our human evaluation can be found in appendix.

## 5.2 CLASSIFICATION PERFORMANCE

Following the settings in LM4CV, we control the bottleneck size to 1 and 2 times of class number and evaluate the classification performance of our proposed method compared with baselines. The results are shown in Table 2 and Table 3. From the results we can observe that our method can outperform state-of-the-art works on concept-based image classification, and both our concept discovery and concept learning method can provide improvement.

We also compare with LaBo following their few-shot settings. In the concept-learning part, to learn the representation for all concepts it needs to fine-tune CLIP with enough training examples. Therefore, we only compare our concept discovery and selection method with LaBo in few-shot settings. For few-shot settings, we also select concepts with few-shot training examples to avoid seeing extra examples.

From the results in Figure 3 we can observe that our discovered concepts consistently outperform LaBo with different bottleneck size in different benchmarks and our concepts can provide significant improvement when limited training examples are available.

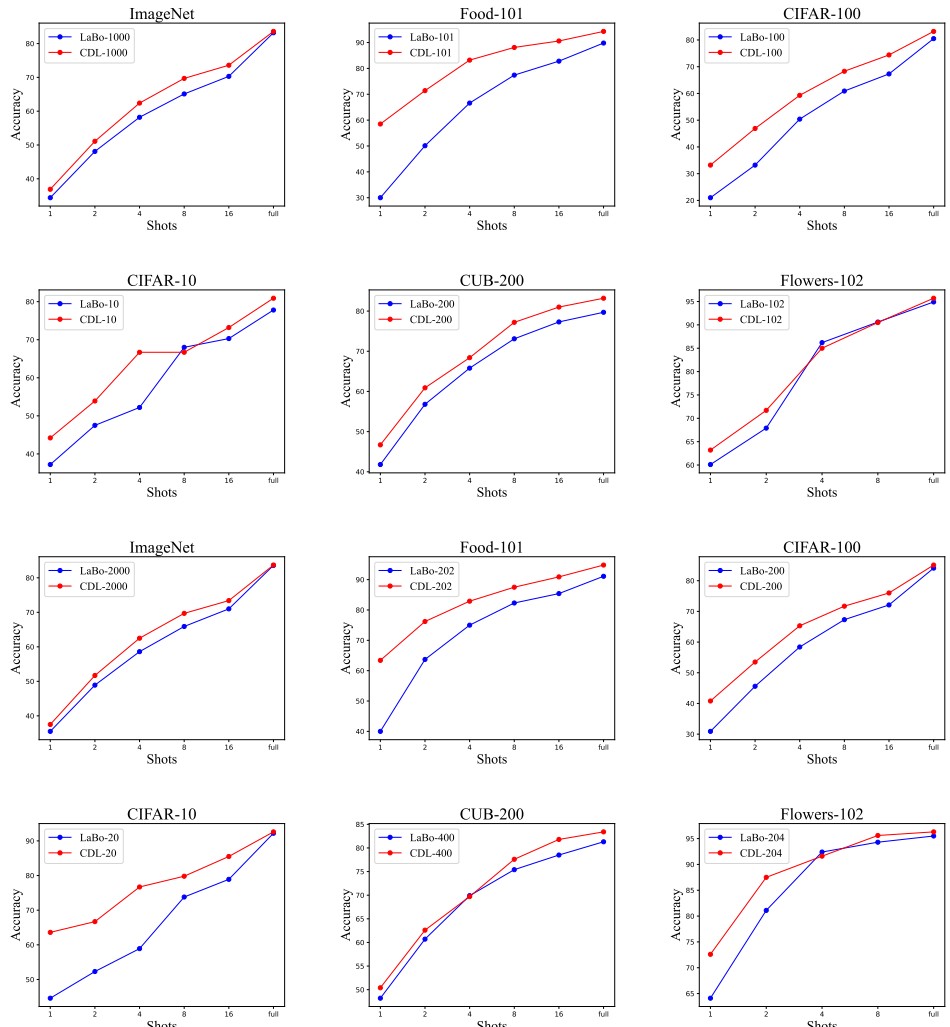

Figure 3: Comparison of test accuracy between LaBo and our method on few shot settings. The number in "Labo-number" and "CDL-number" is the size of the bottleneck (number of concepts).

## 5.3 INTERPRETABILITY OF MULTI-MODAL RECOGNITION

In this subsection, we compare the interpretability of our proposed method and LaBo (Yang et al., 2023b) based on following automatic and human-annotated evaluation metrics.

**Intervention Accuracy** To quantify the interpretability of the CBM on visual concept learning, we introduce the intervention method proposed in Yun et al. (2023). Given a trained CBM, it inputs binary ground-truth concept value (1 if the image contains the corresponding concept otherwise 0) to predict the category. Therefore, high intervention accuracy means that the CBM can learn a accurate concept-category map.

**Factuality** Besides automatic methods, it is also important to judge whether the decision basis of the CBM (the top-weighted concepts) really describe the corresponding categories. Hence, we introduce the metric "Factuality" in LaBo, which evaluate whether the top-$k$ weighted concepts for a category actually appear in the ground truth images of that category. Following their setup, we randomly select 10 categories for each dataset and 10 images from the test dataset for each category. For each concept $c$ in the top-$k$ weighted concepts for each category, we ask human annotators to

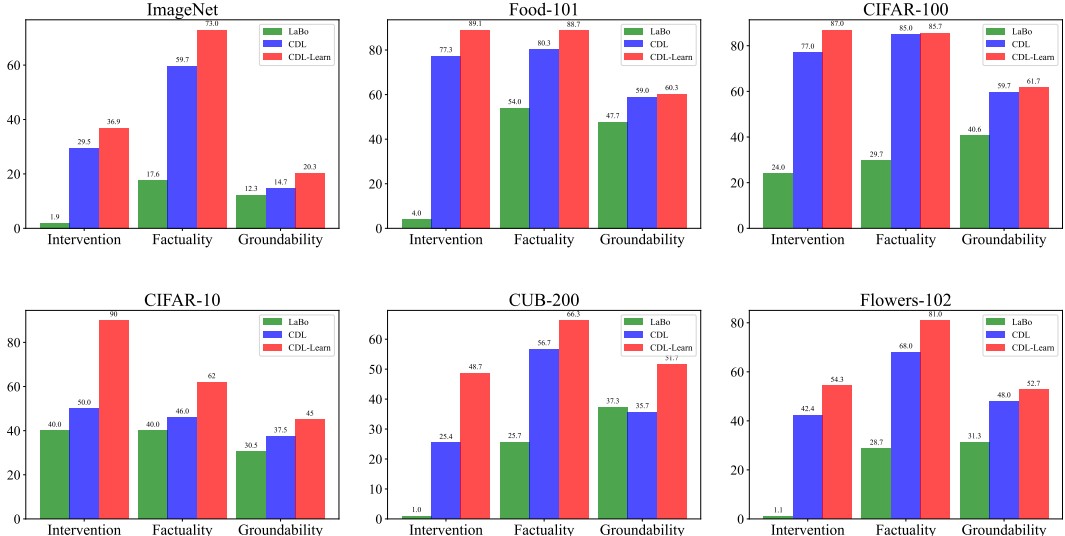

Figure 4: The evaluation of interpretability on different datasets. "CDL" means original CLIP with our discovered concepts and "CDL-learn" means the fine-tuned CLIP with our discovered concepts.

judge whether this concept appears in the images of the corresponding category. The factuality of each concept c is calculated by:

$$Factuality(\text{c}) = \frac{\text{number of images containing c}}{\text{10 ground-truth images of this category}} \quad (4)$$

We represent the factuality of the classification model on a dataset with the mean factuality of all top-k concepts.

**Groundability**    Following LaBo, we also evaluate the vision-language model grounding of the concepts that serve as the decision basis. We utilize the metric "Groundability", which evaluate whether a concept actually appear in the top-10 aligned images ranked by CLIP image-concept similarity score. We use the same human evaluation setting as "Factuality" to evaluate the groundability, that is,

$$Groundability(\text{c}) = \frac{\text{number of images containing c}}{\text{top-10 alignment images of c}} \quad (5)$$

Figure 4 shows the evaluation results of the interpretability of LaBo and our method on different datasets. From the results we can observe that although achieving high classification performance, the LaBo model might not provide enough interpretability because of the low intervention, factuality and groundability of the trained CBM, which is also illustrated in the examples in Figure 1. In the mean time, the results show that our discovered class-agnostic concepts and concept-learning method can provide significant improvement on the interpretability metrics and ameliorate the classification performance. The experimental results prove that our proposed concept discovery and learning framework can build a more factual and groundable multi-modal recognition system.

## 6    CONCLUSION

In this paper, we first dive into the interpretability of VLM-based multi-modal recognition and reveal the limitations of previous works including classname-biased descriptors and non-visual concepts. To overcome these limitations, we proposed a method to automatically discover and select class-agnostic and discriminative visual concepts and fine-tune the CLIP model with a self-supervised concept learning objective. Experimental results on various metrics demonstrated the effectiveness of both concept discovery and learning parts of our approach, with significant improvements in classification accuracy and interpretability. Our work contributes to understanding and improving the concept learning of foundational VLMs.

## 7 REPRODUCIBILITY

We provide the implementation details and hyperparamemters to reproduce our experiments in Section 5.1. We also attach our code for concept discovery and learning in the supplementary meterials. Our code, models, and result files will be publicly released upon acceptance.

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

## A  DETAILS OF DATASETS

The table 4 shows the statistical details of the datasets we choose.

| Dataset | #Class | #Train | #Valid | #Test |
|---------|--------|--------|--------|-------|
| ImageNet | 1000 | 128,1167 | 50,000 | - |
| Food-101 | 101 | 60,600 | 15,150 | 25,250 |
| CIFAR-100 | 100 | 40,000 | 10,000 | 10,000 |
| CIFAR-10 | 10 | 40,000 | 10,000 | 10,000 |
| CUB-200 | 200 | 4,794 | 1,200 | 5,794 |
| Flowers | 102 | 4,093 | 1,633 | 2,463 |

Table 4: Statistical details of datasets. "#Class" means the number of classifications. "#Train", "#Valid", and "#Test" denote the instance numbers of each dataset respectively.

## B  CHOICE OF DIFFERENT PROMPTS

In this section, we discuss the choice of different prompts in concept discovery and compare the performance of concepts discovered with different prompts on the CUB dataset. From the result we can observe that different prompts provide similar performance, which is because the large language model is not sensitive to the prompts and give similar concepts.

## C  PERFORMANCE OF DIFFERENT SIMILARITY THRESHOLD

In this section, we show the performance of our CDL model with different similarity threshold on the CUB dataset. From the results we can observe that the threshold of 0.9 can achieve the best performance.

## D  UNSURPERVISED CLASSIFICATION RESULT WITH DIFFERENT BACKBONES

In this section we compare the unsupervised classification result of our fine-tuned CLIP and previous method (VDES) on different backbones. The comparison with "ViT-L/14" backbone is shown in Sec 4.1. Here we show the comparison with "ViT-B/32" backbone.

## E  HUMAN EVALUATION DETAILS

We hire workers on https://www.mturk.com to conduct human evaluation. In order to make sure the correctness of human annotation, for one data point we ask three human workers to annotate. For the factuality and groundability metric, we randomly sample 10 classes from each dataset and annotate the factuality and groundability of the top-3 concepts of each class. In order to calculate the factuality and groundability, we select 10 images for each concept to annotate. Therefore, we annotate 10,800 data points in total for those two task. For the visual discriminability and classname containing, we conduct annotation on selected 400 concepts of LaBo and our method on the CUB

| Prompts\#Concepts | 200 | 400 |
|-------------------|-----|-----|
| What are useful visual features for distinguishing a {category name} in a photo? | 83.2 | 83.4 |
| What visual features do you use to recognize a {category name} in a photo? | 83.0 | 83.3 |
| What are the identifying features of a {category name} in a photo? | 82.9 | 83.3 |

Table 5: Classification Performance of concepts generated by different prompts on the CUB dataset.

| #Concepts | 200 | 400 |
|---|---|---|
| Threshold = 0.8 | 81.8 | 82.5 |
| Threshold = 0.85 | 82.3 | 82.7 |
| Threshold = 0.9 | **83.2** | **83.4** |
| Threshold = 0.95 | 82.9 | 83.1 |

Table 6: The performance of CDLwith different threshold on the CUB dataset.

| | ImageNet | Food-101 | CIFAR-100 | CIFAR-10 | CUB-200 | Flowers-102 |
|---|---|---|---|---|---|---|
| CLIP + Name | 58.5 | 79.3 | 63.5 | 89.0 | 52.0 | 65.9 |
| CLIP + Name w/ Concept | **63.0** | **83.6** | 64.7 | 90.3 | 52.6 | 66.1 |
| CLIP + Concept | 16.2 | 2.5 | 22.8 | 59.4 | 3.2 | 4.6 |
| CLIP + Name w/ Random Concept | 61.2 | 80.4 | 63.3 | 90.1 | 52.6 | 66.3 |
| CDL + Concept | 62.7 | 82.0 | **65.2** | **90.7** | **53.9** | **67.4** |

Table 7: The unsupervised classification results of the original and our fine-tuned CLIP model with different prompts. "Name" corresponds to the simple prompt "A photo of a class name". "Name w/ Concept" denotes the prompts in the previous work (Menon & Vondrick, 2022), which are like "A photo of a class name, which has "concept". "Concept" corresponds to the pure concept. "Name w/ Random Concept" means that we replace the correct concept with random concepts. The large gap between "Name w/ Concept" and "Concept" and the small gap between "Name w/ Random Concept" and "Name w/ Concept" mean that the class names instead of the descriptive features in the prompts make the main contribution to the decision of the CLIP model. "CDL + Concept" means the prediction of our fine-tuned CLIP model with class-agnostic concepts.

dataset. Hence we annotate 1,600 data points for those two task. We pay the human workers $0.05 each data point. The total cost of human annotation is $1,860. In the annotation, we randomly shuffle the order of instances to remove possible biases.

In order to validate the effectiveness of our human evaluation, we calculate the pairwise annotator agreement score following previous work Yang et al. (2023b). The average pairwise annotator agreement propotion on all datasets is 69.2%, which is comparable with the 69.8% propotion in the previous work.

We conduct Students' T-test to evaluate the statistical significance of the human evaluation results. We set the threshold of p-value to be 0.05 following previous works. When p-value is lower than 0.05, the null hypothesis is rejected and out method performs significantly better than the baseline method. From the results we can observe that both our concept learning and concept discovery method significantly outperform the baseline methods regarding the intervention, factuality and groundability metrics.

We show some examples about the interface of our human annotation. In the annotation platform, the workers can see an image and is asked to select whether the given concept describes the image.

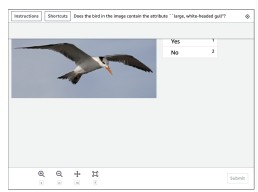 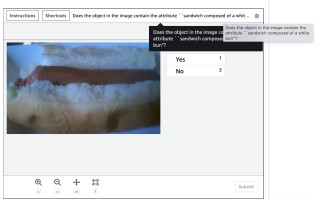 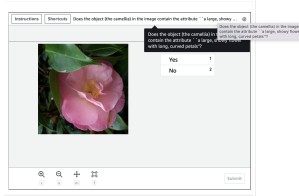

Figure 5: Examples of the annotation interface.

| Dataset | Method | Intervention | | Factuality | | Groundability | |
|---|---|---|---|---|---|---|---|
| | | p-value | significance | p-value | significance | p-value | significance |
| ImageNet | CLIP + CDL v.s. CLIP + LaBo | 2.6e-22 | ✓ | 4.3e-29 | ✓ | 0.34 | × |
| | CDL + CDL v.s. CLIP + CDL | 5.6e-2 | × | 5.2e-4 | ✓ | 8.4e-2 | × |
| Food-101 | CLIP + CDL v.s. CLIP + LaBo | 1.3e-106 | ✓ | 5.5e-12 | ✓ | 5.3e-3 | ✓ |
| | CDL + CDL v.s. CLIP + CDL | 8.5e-5 | ✓ | 3.4e-3 | ✓ | 0.80 | × |
| CIFAR-100 | CLIP + CDL v.s. CLIP + LaBo | 9.2e-45 | ✓ | 1.1e-50 | ✓ | 1.8e-6 | ✓ |
| | CDL + CDL v.s. CLIP + CDL | 1.3e-3 | ✓ | 0.82 | × | 0.62 | × |
| CIFAR-10 | CLIP + CDL v.s. CLIP + LaBo | 1.3e-2 | × | 0.14 | × | 7.0e-2 | × |
| | CDL + CDL v.s. CLIP + CDL | 2.7e-29 | ✓ | 7.8e-5 | ✓ | 5.6e-2 | × |
| CUB-200 | CLIP + CDL v.s. CLIP + LaBo | 8.9e-20 | ✓ | 2.8e-15 | ✓ | 0.73 | × |
| | CDL + CDL v.s. CLIP + CDL | 2.0e-9 | ✓ | 1.9e-2 | ✓ | 7.2e-5 | ✓ |
| Flowers-102 | CLIP + CDL v.s. CLIP + LaBo | 1.5e-39 | ✓ | 1.2e-23 | ✓ | 1.8e-5 | ✓ |
| | CDL + CDL v.s. CLIP + CDL | 4.2e-3 | ✓ | 2.4e-4 | ✓ | 0.25 | × |

Table 8: The statistical significance of the human evaluation results.

| Category | Concept Selected | Concept Excluded |
|---|---|---|
| Giant Panda | black patches around eyes
large, round head
black fur on ears | a rare animal
popular in zoo |
| Black-footed Albatross | black and white
a long, hooked bill
long, narrow wings | found in North America
dive to depths of over 30 meters |
| Grey Whale | long, curved mouth
dark grey or black
white patches on the skin | large marine mammal
long-distance magrition |

Table 9: The examples of selected and excluded concepts by our Mutual Information based concept selection method

## F  EXAMPLES OF MUTUAL INFORMATION BASED CONCEPT SELECTION

In this section we showcase some examples of the concepts selected by our Mutual Information based method. From the examples we can see that our method can effectively select visually discriminative concepts and exclude non-visual ones.

## G  EXAMPLES OF CONCEPT-BASED MULTI-MODAL RECOGNITION

In this section we show some examples of different concept-based image classification methods. From the examples in Figure 6 we can observe that previous works suffer from class-conditional and non-visual concepts, while our method can learn interpretable concept-class map based on class-agnostic concepts.

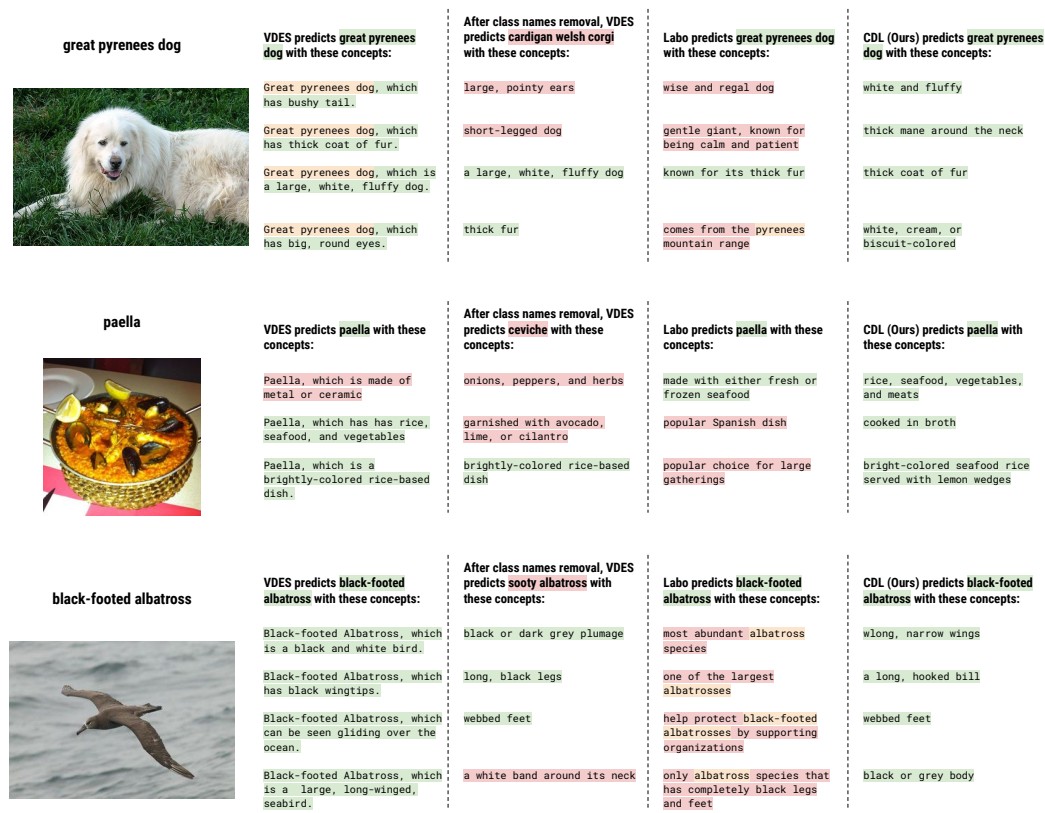

Figure 6: Examples of how different models conduct image classification based on the concepts. Correct predictions and concepts are in green , while wrong concepts and non-visual concepts are in red . Though VDES (Menon & Vondrick, 2022) and LaBo (Yang et al., 2023b) can both classify the image correctly and the concepts are mostly correlated with the class names (highlighted in orange ). After the removal of class name in VDES, we observe that VDES classifies this image as ring tailed lemus and correlate the image with irrelevant concepts. Our proposed method (CDL) can predict *giant panda* correctly based on the class-agnostic concepts.

