# OpenReview forum: "Interpretable Concept Discovery and Learning from Pretrained Vision-Language Models"
_ICLR.cc/2024/Conference — Submitted to ICLR 2024_

### Official Review · Reviewer_PAjz · 2023-10-30

**Soundness:** 3 good
**Presentation:** 3 good
**Contribution:** 2 fair
**Rating:** 5
**Confidence:** 3

**Summary:**

This paper proposes a concept discovery and learning (CDL) framework aiming to better align the semantic representation for a category label and its relevant attribute labels. It is inspired by an observation that in current visual-language models (VLM), there are inconsistencies between class labels and the attributes.

To mitigate such inconsistency, given a category label (e.g., a panda), CDL queries a large language model (LLM) about useful visual features that helps recognition of that category. This process results in a concept-class corelation matrix. A mutual information inspired method is then used to obtain the most informative concepts for a specific class. In the end, each class will correspond to a 0-1 vector indicating whether a concept is useful for that class. The obtained concept vectors are then used to finetune the last projection layer of CLIP. Through this process, the paper claims that CDL helps visual-language model learn a better correspondence between class labels and the attributes.

Experimental studies on ImageNet, Food-101, CIFAR-100, and a few smaller datasets were conducted to show the efficacy of the proposed method. A human study on the interpretability of the CDL framework is also provided.

**Strengths:**

- The paper is technically and intuitively correct.
- Extensive experiments on different image datasets and human study about interpretability are conducted.

**Weaknesses:**

- Table 1 is used as a support of the paper's claim about the misalignment of concepts and classes in CLIP. However, there is not a similar table for CLIP+CDL using the same evaluation to show that it does improve the alignment between classes and concepts. Current experiments only show that VLM+CDL becomes a good concept detector but does not say that it effectively relates concepts with classes.
- Mutual information-based concept selection is correct but not new. Moreover, I do not see the necessity of introducing MI here. The paper should better rationalise the choice of this criterion.
    * How does using MI differ from using the normalised cooccurrence frequency between categories and classes?
    * Is there any example showing the concepts selected for different classes?
- Missing details about the similarity threshold ``$\texttt{th}$'' (page 5). There is no further discussion on it except in Section 3.2.
    * What is the typical value for it?
    * Is the parameter fixed for all the datasets?
    * Is the framework sensitive to this parameter?
- Missing details about human study.
    * There are no error bars in Figure 4, making it difficult to tell how well the human annotators agree and how significantly CDL improved over prior methods. Since each data point has been annotated by three human annotators (as described in the appendix), there should be at least an error bar for each result in the figure.
    * Moreover, there is no information about the interface viewed by the annotators. For example, Are samples fairly displayed to the annotators, removing the possible bias caused by the order or the position of display?
- Clarity: what is the "CDL+CDL" setting in Table 2?
- Typo: "directlyed" in page 2.

**Questions:**

Please see my points above. As a summary, I expect answers regarding:
1. Does VLM trained with CDL really better models the relation between concepts and classes?
2. MI for concept selection is good, but why bother using it here? Any ablation study about it?
3. Missing details about model design, for example the similarity threshold $\texttt{th}$.
4. Missing details about human study, for example the error bars.

---

> ### Author Response · Authors · 2023-11-22
>
> We thank the reviewer PAjz for the constructive suggestions and insightful questions!
>
> #### Q1: The improvement of CDL on alignment between classes and concepts
>
> In order to demonstrate the ability of our CDL model to align concepts with classes, we add the experiment to compare our CDL model with VDES on zero-shot classification in Table 1 (last row, “CDL + Concept”). The results show that our fine-tuned CLIP model can achieve competitive performance while using only class-agnostic concepts. Notably, when VDES is modified to rely only on its concepts, the performance drops significantly (the row “CLIP + Concept”). The results illustrate that our CDL method can learn correct concept and class association and provide interpretability on how VLMs utilize primitive concepts for recognition, and that CDL can indeed improve alignment between classes and concepts.
>
> #### Q2: The necessity of Mutual Information
>
> As illustrated in Figure 1 (fifth column, “Labo”), we observe that previous methods include many non-visual concepts (e.g. “keep the panda warm”) in their discovered concepts. We hypothesize that the selection of such concepts would make the concept-based representation less visually discriminative when the concept vocabulary size is fixed. Hence, we introduce Mutual Information to select visually discriminative concepts.
>
> The Mutual Information between the image-concept similarity and the ground-truth containment of the image for the concept can measure how well the concept can be recognized by the VLM and can select **visually salient** concepts. If we use other methods like normalized co-occurrence frequency between categories and classes to select concepts, it still cannot distinguish visual and non-visual concepts. For example, the non-visual concepts like “long life span” and “a very rare animal” also usually co-occur with “giant panda”, and we need to introduce the vision information to filter them.
>
> To better showcase the effectiveness of the Mutual Information based concept selection, we add some examples of the concepts selected and excluded by the method in Table 9 of the Appendix. From the examples we can see that our method can effectively select visually discriminative concepts and exclude non-visual ones.
>
> #### Q3: Details about the threshold
>
> We fully agree that adding more details of the similarity threshold is necessary. The similarity threshold is 0.9 for our current method. Intuitively, a lower threshold will select fewer concepts and a higher threshold will include more similar concepts. We add the details of the similarity threshold selection in the experiment section and show the performance of different thresholds on the CUB dataset in Table 6 of the Appendix. The results show that the performance of the CDL model is reasonably robust against the choice of the threshold.
>
> #### Q4: Details about human study
>
> Thank you for your suggestions on the presentation of the human study.We agree that more details about human study need to be included.
> - Error bars about human evaluation: we conduct the Students’ T-test to evaluate the statistical significance of the human evaluation results. The results are shown in Table 8 of the Appendix. From the results we can observe that both our concept learning and concept discovery method significantly outperform the baseline methods regarding the intervention, factuality and groundability metrics. We also report the pairwise annotator agreement score of our human evaluation to validate the effectiveness of our human evaluation.
> - Potential biases in the annotation process: for the annotation, we randomly shuffled the order of instances to remove possible biases. We add the examples of human annotation interfaces in Figure 5 of the Appendix. Since the annotator receives one image each time, the position will not bring biases to the result.
>
> #### Q5: Clarifying the notations in the tables
>
>  For the notations `A+B` in Table 2, `A` represents the CLIP model and `B` represents the concept discovery method. `CLIP+LaBo` represents the original CLIP model with LaBo concepts. `CLIP+CDL` represents the original CLIP model with our discovered concepts. `CDL+CDL` represents our fine-tuned CLIP model with our discovered concepts.
>
> #### Q6: Typo
> Thank you for pointing out our typos. We have corrected them in the revised version of our paper.

---

### Official Review · Reviewer_8mWh · 2023-11-01

**Soundness:** 3 good
**Presentation:** 2 fair
**Contribution:** 2 fair
**Rating:** 5
**Confidence:** 3

**Summary:**

The study delves into Vision-Language Models (VLMs), specifically models like CLIP, and their proficiency in discerning and utilizing visual concepts such as colors and shapes for multi-modal recognition. Past research offers mixed views: some findings suggest VLMs might lack in interpretability, while others indicate that concept-based text prompts can enhance recognition and offer some degree of interpretability. This paper attributes these discrepancies to varied concept definitions and prompting methods among prior works. To address this, a novel framework is introduced to extract and learn interpretable, class-agnostic visual concepts from pretrained VLMs. These concepts are selected based on their visual distinctiveness, evaluated through mutual information between the images and the concepts. A self-supervised approach is then proposed to refine the VLM's recognition capabilities of these concepts. Results, supported by extensive quantitative and human evaluations, confirm that this approach not only bolsters interpretability but also enhances recognition accuracy.

**Strengths:**

The idea is reasonable and contributes to the interpretability. Experimental results also support that the proposed approach outperforms the baselines.

**Weaknesses:**

1. This paper found out that the classification accuracy drops significantly when the input prompt for the text encoder is without the class name. However, not using the class names may not be a large problem. The key concept of using CLIP is based on the contrastive learning between image-text pairs, which is powerful. As such, it is reasonable that the class names associated with the descriptions improvement the classification. Moreover, the model with class names can provide the correct description, e.g., the examples in Fig. 1.
2. The method for alleviating the problem is to use different prompts, e.g., “What are useful visual features for distinguishing a {category name} in a photo?,” which is heuristic. Another alternatives should be considered and compared.
3. The experiments should follow the similar setting with previous work. It is suggested to compare with VDES (and following works) and show the results with different backbones.
4. The references are out-of-date. Only two papers published in 2023 are cited. Moreover, VDES is published in ICLR 2023, while the reference in the paper is still an arxiv paper.

**Questions:**

It is suggested to provide a little bit more details about previous work (Menon & Vondrick, 2022) to improve the readability.

---

> ### Author Response · Authors · 2023-11-22
>
> We appreciate the helpful comments and suggestions from reviewer 8mWh!
>
> #### Q1: About using the class name in the concepts
>
> This is a great point! We agree with the reviewer that class names are helpful for classification and can be used to improve empirical performance on zero-shot classification benchmarks. We would like to clarify that our main point is that the use of class names may bias the analysis on whether VLMs (e.g. CLIP) are able to learn and use concepts (e.g. black patches around the eyes). Specifically, the concepts retrieved by VDES may be correct for two reasons: First, CLIP captures the concept and uses it for zero-shot classification. Alternatively, CLIP uses the class name for zero-shot classification, while the concepts are only retrieved because GPT-3 indicates the concepts are correlated with the class names.
>
> To better illustrate the point, we revise Figure 1 and show a scenario where VDES is applied on class names with randomly shuffled concepts (e.g. giant panda, which has black wings), and we can observe that the class names are correctly retrieved despite having completely irrelevant concepts. This observation indicates that the class names instead of the descriptive concepts make the decisive contribution to the recognition. We further validate this observation in Table 1 (fourth row, `CLIP + Name w/ Random Concept`), which shows that pairing the class names with randomly shuffled irrelevant concepts do not hurt the zero-shot classification performance.
>
> #### Q2: Other alternative prompts for concept discovery
>
> We fully agree that the comparison between different prompts is important to show. We have conducted comparisons between different alternatives for the prompts for concept discovery. The results are in Table 5 of the Appendix. From the results we can observe that different prompts provide similar performance, which demonstrates the robustness of our observations.
>
> #### Q3: Comparison with VDES
>
> We have added this suggested experiment in the revised Table 1 (last row). The row `CLIP + Name w/ Concept` represents the framework of VDES and the row `CDL + Concept` represents our fine-tuned CLIP with the discovered concepts. The results show that our fine-tuned CLIP model can achieve competitive performance while using only class-agnostic concepts. Notably, when VDES is modified to rely only on its concepts, the performance drops significantly (the row `CLIP + Concept`). The results illustrate that our CDL method can learn correct concept and class association and provide interpretability on how VLMs utilize primitive concepts for recognition.
>
> #### Q4: Out-of-date reference
>
> Thank you for pointing it out. We have referred to more recently published works in the related work section and corrected the wrong references. Now we have 10 references published in 2023.
>
> #### Q5: Details about VDES
> Thank you for your valuable advice on our presentation. We have added more details for VDES in Section 4.1 of our revision.

---

### Official Review · Reviewer_JkAA · 2023-11-10

**Soundness:** 2 fair
**Presentation:** 1 poor
**Contribution:** 1 poor
**Rating:** 3
**Confidence:** 4

**Summary:**

This paper proposed a  framework to jointly discover and learn interpretable visual concepts from pretrained VLMs.  The authors claim that the discovered concepts are class-agnostic, and selected based on the visual discriminability measured by mutual information between images and concepts.
Besides this, the authors propose a self-supervised framework to adapt the CLIP models to recognize the discovered concepts. Experiments on several datasets show these concepts are helpful for understanding CLIP models.

**Strengths:**

+ easy to follow and implement
+ clear figures for readers to understand
+ present experiments on multiple datasets.

**Weaknesses:**

The technical novelty. The overall framework aims to decompose an existing class name into basic visual concepts and then compose them into a semantic verb for final supervision. This structure seems to be trivial and does not provide us with many insights. The ranking for concepts simply borrows the definition of mutual information as Eq.(3). The reviewers doubt the overall novelty of this framework and the contributions seem to be weak.

The experimental results somehow do not support the overall idea. From Tab.1 using concepts provides little performance improvements. In Tab.2, boosting the concepts from 1000 to 2000 provides only 0.1% improvements on the public imageNet dataset. The experimental results do not fully support the proposed contributions.

The overall presentations and organizations are not well exhibited. The paper is somehow not easy to follow the key idea and many insights behind the simple language concepts are not clearly explored. The reviewers suggest the authors further explore the semantic concepts or some hidden contextual information rather than this simple architecture.

**Questions:**

Please refer to the weakness section above. Considering its overall quality, I tend to give a negative score.

---

> ### Author Response · Authors · 2023-11-22
>
> We would like to thank reviewer JkAA for their constructive feedback! We would like to clarify the significance of our contributions, and how the results support our contributions.
>
> #### Q1: On technical novelty and contributions.
>
> We would like to highlight that our work is motivated by a line of research aiming to improve the interpretability and explainability of vision-language foundation models (VLMs). We aim to answer the fundamental research question: Do VLMs learn to represent images in terms of interpretable, more atomic visual concepts, such as colors and shapes? Knowing the answer to this question would not only help us better understand the potentials and limitations of the existing VLMs, but have important applications for compositional generalization (e.g. recognizing a purple banana despite only observing purple eggplants and yellow bananas during training), and performing test-time intervention (correcting the incorrect concepts and thus making more accurate final predictions).
>
> - Structure is trivial: We respectfully disagree. We adopt the same model structure as prior work, such as CBM (ICML 2020), CompMap (TMLR 2023), and LaBo (CVPR 2023). We assume that composite concepts (e.g. purple bananas) can be composed from primitive concepts (e.g. purple and bananas) using a linear model (i.e. the concept bottleneck classifier), which naturally allows us to inspect how the concepts are utilized to infer the composite concepts, when models are trained. We believe our assumption generalizes to real world scenarios.
> - Not many insights: Although we are not the first work that attempts to analyze if interpretable primitive concepts are captured by VLMs, we are the first one to reveal the limitations of previous analysis, namely the class-name bias. In the revised Figure 1, we show that existing concept-based methods, such as VDES (ICLR 2023), rely on class names as opposed to concepts to make decisions. When the class name is removed (third column from the left), the predicted class is wrong. When the class name is retained but the concepts are randomly shuffled (fourth column from the left), the predicted class remains correct but the retrieved concepts are all wrong. This observation is further validated in Table 1 (3rd and 4th rows). The class-name bias prevents us from properly understanding if the concepts are encoded by pre-trained VLMs, and we designed a framework of concept discovery and learning to address these issues.
>
> Inspired by our observations above, we propose to discover class-agnostic (hence not class-name-biased) and visually discriminative concepts with the help of LLM and our proposed Mutual Information concept selection process. Compared to CBM and CompMap, our concepts are discovered as opposed to manually designed. Compared to LaBo, our concepts are chosen to be compact and visually discriminative, leading to better interpretability and higher recognition accuracy. Moreover, we propose a novel and efficient self-supervised concept learning framework to learn the association between concepts and classes, which achieves much better interpretability according to the Intervention, Factuality and Groundability metrics (see Section 5.3).

---

> ### Author Response · Authors · 2023-11-22
>
> #### Q2: From Tab.1 using concepts provides little performance improvements
>
> We would like to clarify that Table 1 aims to illustrate that the class names play the crucial role for the zero-shot classification accuracy across benchmarks. Using concepts (as performed by VDES) does provide moderate performance gains, but our point is to illustrate that the gain is not obtained via better understanding of concepts. When we remove the class names (the `CLIP + Concept` row) the performance drops substantially. When we randomly shuffle the concepts (the `CLIP + Name w/ Random Concept` row) the performance remains despite wrong concepts being utilized. We visualize such behaviors in Figure 1.
>
> However, as shown in the last row of the revised Table 1, after our proposed concept discovery and learning, the concept-only approach is able to achieve competitive performance on zero-shot classification (outperforms the `CLIP+Concept` row significantly), where the concepts are used for the “right” reasons (unlike the `CLIP + Name w/ Concept` row). Compared with the CLIP baseline (first row), our results demonstrate that the correct way of using concepts not only provides interpretability, but also improves the zero-shot classification accuracy, especially for ImageNet, CUB, and Flowers.
>
> #### Q3: In Tab.2, boosting the concepts from 1000 to 2000 provides only 0.1\% improvements on the public imageNet dataset.
>
> Thank you for pointing this out! This is exactly the expected behavior since we view learning a compact concept space that is effective for visual recognition as one of the advantages of our framework. We have demonstrated that even when we use a small concept space (equal to the number of classes in the target dataset), our approach still performs competitively.
>
> #### Q4: The overall presentations and organizations are not well exhibited.
>
> We fully agree. We would like to incorporate our responses above and improve the overall presentation of our paper. Thank you so much for your feedback, and we look forward to your more detailed comments on the places we should improve!
>
> Although we believe our model architecture is well grounded and motivated, and our insights to be novel, we agree with the reviewer and plan to explore alternative architectures to properly understand the potential limitations of the concept bottleneck architecture.

---

### Author Response · Authors · 2023-11-22
**Update of the paper**

We would like to thank all reviewers for their constructive and detailed feedback! We have uploaded a revised version to reflect the suggested changes (marked in red). Notably, we updated Figure 1 to illustrate that existing concept-based zero-shot classifiers, although appear to provide interpretable concept descriptions (second column), actually relies on the class names to make predictions. This is supported by our example in the fourth column, that when we randomly shuffle the concepts with the class names, the correct class names are selected with the irrelevant concepts.

To further illustrate how our proposed concept discovery and learning approach would resolve this issue, we conduct new experiments to evaluate our fine-tuned model on the zero-shot classification benchmark (Table 1, last row). We can observe that our fine-tuned model can achieve competitive performance while using only **class-agnostic** concepts, without relying on the class names. This experiment confirms that our model can learn correct associations between concepts and classes and provide concept-level interpretability.

---

### Meta-Review · Area_Chair_Hgws · 2023-12-05

**Metareview:**

(a) Summary of Scientific Claims and Findings:
The paper presents a framework for discovering and learning interpretable visual concepts from pretrained Vision-Language Models (VLMs), particularly focusing on the CLIP model. It proposes a self-supervised approach to adapt these models to recognize the concepts identified. The concepts are class-agnostic and are selected based on visual discriminability measured by mutual information between images and concepts. The paper asserts that these concepts enhance the understanding of CLIP models, supported by experiments on various datasets.

(b) Strengths of the Paper:
1. The framework is straightforward and easy to implement.
2. The paper contributes to the interpretability of VLMs.

(c) Weaknesses of the Paper:
1.Lack of technical novelty: The approach seems trivial and lacks significant insights.
2.Inadequate experimental support: Minor performance improvements are observed, questioning the effectiveness of the proposed method.
3.Inconsistencies in results: The paper doesn't adequately demonstrate the effectiveness of the CDL framework in aligning concepts with classes.
4.Concept selection methodology: The use of mutual information is not novel and lacks justification.

**Justification For Why Not Higher Score:**

The paper does not receive a higher score due to its limited technical novelty, lack of compelling experimental evidence, and presentation issues. The conceptual framework does not significantly advance the field, and the results fail to convincingly support the claimed benefits. Moreover, the paper suffers from organizational flaws, making it challenging to grasp the full scope and impact of the proposed method.

**Justification For Why Not Lower Score:**

Despite its shortcomings, the paper offers a reasonable approach towards improving interpretability in VLMs and conducts extensive experiments. These aspects, along with the clarity of figures and the idea's implementation ease, warrant consideration. However, improvements in novelty, experimental validation, and presentation are crucial for a higher evaluation.

---

### Decision · Program_Chairs · 2024-01-16

Reject